# Intrinsic Network Changes in Bilateral Tinnitus Patients with Cognitive Impairment: A Resting-State Functional MRI Study

**DOI:** 10.3390/brainsci12081049

**Published:** 2022-08-08

**Authors:** Wei Li, Xiaobo Ma, Qian Wang, Xueying He, Xiaoxia Qu, Lirong Zhang, Lanyue Chen, Zhaohui Liu

**Affiliations:** 1Department of Radiology, Capital Medical University, Beijing Tongren Hospital, Beijing 100730, China; 2Department of Otolaryngology Head and Neck Surgery, Capital Medical University, Beijing Tongren Hospital, Beijing 100730, China; 3Department of Radiology, Medical School of Nanjing University, Afliated Drum Tower Hospital, Nanjing 210008, China

**Keywords:** tinnitus, hearing loss, cognitive impairment, resting state functional magnetic resonance imaging, independent component analysis, functional connectivity

## Abstract

Previous studies have found a link between tinnitus and cognitive impairment, even leading to dementia. However, the mechanisms underlying this association are not clear. The purpose of this study was to explore intrinsic network changes in tinnitus and hearing loss patients with cognitive disorders. We included 17 individuals with bilateral idiopathic tinnitus, hearing loss, and cognitive impairment (PA) and 21 healthy controls. We identified resting-state networks (RSNs) and measured intra-network functional connectivity (FC) values via independent component analysis (ICA). We also evaluated correlations between RSNs and clinical characteristics. Compared with the healthy controls, the PA group showed decreased connectivity within the ventral attention network, dorsal attention network (DAN), visual network, left frontoparietal network, right frontoparietal network, sensorimotor network, and increased connectivity within the executive control network. MoCA (Montreal Cognitive Assessment) scores were negatively correlated with the FC values for left calcarine within the DAN. We identified abnormal intrinsic connectivity in several brain networks, mainly involving cognitive control, vision, sensorimotor function, and the cerebellum, in tinnitus patients with cognitive impairment. It may be possible to use the FC strength of the left calcarine within the DAN as an imaging marker to predict cognitive impairment in tinnitus patients.

## 1. Introduction

Tinnitus is an auditory phantom sensation, which affects approximately 15% of the general population [1]. The most common form is idiopathic tinnitus, which refers to tinnitus that has no cause other than sensorineural hearing loss [2]. Tinnitus can negatively influence human health in a large number of ways. For instance, long-term tinnitus can lead to disturbed sleep, somatization, anxiety, depression, and even suicide [3]. Tinnitus patients are generally eager to seek clinical help. Unfortunately, because the underlying mechanisms are unclear, few individualized and effective treatment options exist.

According to previous studies, tinnitus can be caused by damage to the cochlea. However, approximately 5% of patients with tinnitus have normal hearing and show no evidence of cochlear damage [4]. Furthermore, tinnitus symptoms remain even after cochlear lesions have healed [1]. Therefore, tinnitus is not solely caused by peripheral damage [5]. A growing number of scholars believe that the central nervous system plays an important role in the pathophysiology of tinnitus. For instance, Sedley [6] proposed that central nervous system alterations are the underlying cause of tinnitus, and that brain network modulation underlies the development, progression, and regulation of persistent tinnitus. Numerous functional imaging studies have described abnormal auditory and non-auditory network function in tinnitus patients, but the data vary widely. Some authors have proposed that limbic and auditory brain areas interact at the thalamic level for noise cancellation, leading to tinnitus perception [7,8], while others have supposed that tinnitus arises from an expectation mismatch within the auditory system [9]. One tinnitus model suggests that tinnitus results from abnormal corticostriatal gating mechanisms [10]. Some researchers have suggested that tinnitus may be the result of maladaptation of the auditory nervous system [11]. The auditory network is the part of the cerebral cortex located in the temporal lobe that performs the function of processing auditory information and is the core of the auditory center [12]. However, no significant differences in auditory network functional connectivity (FC) were found between a tinnitus group and a control group [11,13]. Schmidt [14] and De Ridder [15] both concluded that the variations in the data are related to the heterogeneity of the tinnitus participants, including the side of tinnitus, presence or absence of hearing loss, and level of cognitive impairment.

Recent studies have verified that tinnitus can lead to cognitive impairment and even dementia [16]. Behavioral studies have detected poor cognitive function in tinnitus patients, including changes in working memory, attentional executive control, and processing speed [17,18,19]. Furthermore, some recent functional imaging studies of tinnitus patients have identified structural and functional changes in brain areas associated with cognitive impairment. These changes include hyperactivity in the hippocampal and parahippocampal gyri, the anterior cingulate cortex, and the dorsolateral prefrontal cortex, which are associated with working memory, attention, and emotion, respectively [20,21,22]. Several studies have found that tinnitus patients show abnormal functional connectivity in the limbic system, the default mode network (DMN), the dorsal attention network (DAN), and the visual network, which are associated with attention, emotion, and other cognitive functions [23,24]. However, few studies have examined the mechanisms underlying tinnitus-induced cognitive impairment. Additionally, like tinnitus, hearing loss is a causative factor for cognitive impairment [25], and tinnitus is accompanied by hearing loss in up to 90% of cases [26]. Unfortunately, the contributions of tinnitus and hearing loss to cognitive impairment have not yet been characterized [16].

Resting-state functional magnetic resonance imaging (rs-fMRI) requires minimal patient compliance and is relatively easy to implement in clinical studies. Resting networks are constructed based on FC in long fMRI time series. Independent component analysis (ICA), as a whole brain network functional connection analysis from rs-fMRI, allows for hypothesis-free and observer-independent measures of intrinsic connectivity patterns [27]. Because tinnitus involves multiple brain networks, we used ICA to examine the mechanisms of tinnitus and tinnitus-induced cognitive impairment with the goal of characterizing whole brain intranet changes in FC.

Given the heterogeneity of tinnitus cases, we focused on changes in intrinsic connectivity patterns in bilateral tinnitus patients with hearing loss and cognitive impairment. On the basis of the above studies on tinnitus, we hypothesized that these patients would have abnormal brain networks.

## 2. Methods

### 2.1. Participants

The experimental procedures were approved by the Institutional Review Board of Beijing Tongren Hospital, Capital Medical University (No.TRECKY2013-KS-23). Written informed consent was obtained from all participants before the study began.

We recruited 17 bilateral tinnitus and hearing loss patients from Beijing Tongren Hospital from May 2020 to May 2021. All patients underwent the Montreal Cognitive Assessment (MoCA), and scored < 26. The inclusion criteria for patients were as follows: (1) age over 18; (2) right-handed and ethnically Han Chinese; (3) bilateral idiopathic tinnitus; (4) bilateral sensorineural hearing loss above 25 dB HL. The exclusion criteria were: (1) a history of associated ear diseases such as hyperacusis, otosclerosis, and Ménière’s disease; (2) preexisting mental illness; (3) a history of severe alcoholism or smoking; (4) a history of associated brain diseases confirmed by conventional MRI; (5) insufficient quality of the acquired images.

We enrolled healthy controls (HC) who met the following criteria: (1) right-handed and ethnically Han Chinese; (2) normal hearing verified by audiological examination and no ear diseases; (3) Montreal Cognitive Assessment score ≥ 26; (4) no deformities of the brain parenchyma on cranial MRI; (5) absence of psychiatric diseases. We recruited 21 HCs. Table 1 shows the demographic and clinical data for all participants. No significant differences were observed in sex, age, or education between groups (*p* > 0.05).

### 2.2. Audiological and Psychoacoustic Evaluation

All patients underwent pure-tone audiometry testing (PTA), the tinnitus matching test, the Tinnitus Handicap Inventory (THI), the Montreal Cognitive Assessment (MoCA), the Beck Depression Inventory (BDI), and the State Trait Anxiety Inventory (STAI). 

We used PTA to evaluate hearing status in an acoustically isolated booth. A threshold value greater than 25 dB at any of the six frequencies (0.25, 0.5, 1, 2, 4, and 8 kHz) is considered to reflect hearing loss [28]. The loudness and pitch of tinnitus were determined using the Tinnitus Matching Test, in which patients choose the frequency and loudness of a pure tone that matched their tinnitus [29]. Additionally, we administered the THI to estimate the severity of tinnitus in terms of functionality, emotion, and severity [30]. We used the MoCA to assess the presence of cognitive impairment, with an optimal cut-off value of 26 [31]. The BDI and STAI were used to measure symptoms of depression and anxiety. A summed score ≥ 10 on the BDI is considered to reflect the presence of depression [32]. The STAI includes the State Anxiety Inventory (S-AI) and the Trait Anxiety Inventory (T-AI). While the S-AI examines state anxiety depending on specific circumstances, the T-AI measures trait anxiety and anxious tendencies regardless of situation. For the S-AI and T-AI, scores ≤ 52 and ≤54 are seen to reflect normal anxious tendencies, respectively [33].

### 2.3. Image Acquisition

The images were acquired using a 3.0T MRI scanner (Discovery MR750, General Electric Medical Systems, Milwaukee, WI, USA). A matched 8-channel phase-array head coil was used with earplugs and foam pads to reduce scanner noise and head-motion, respectively. During scanning, all subjects were asked to lie flat on the examination bed with their eyes closed, try to stay still, and not think of anything in particular. The scan images were parallel to the anterior-posterior commissure line and covered the whole brain. The sagittal three-dimensional T1-weighted sequences had the following parameters: repetition time/echo time = 8.8/3.5 ms; field of view = 240 × 240 mm^2^; matrix = 256 × 256; slice thickness = 1.0 mm, no gap; flip angle = 13°; 176 slices. Resting-state fMRI data were obtained using an echo planar imaging (EPI) pulse sequence with the following parameters: repetition time/echo = 2000/35 ms; field of view = 240 × 240 mm^2^; matrix = 64 × 64; slice thickness = 3.00 mm, 1.00 mm gap; flip angle = 90°; 180 volumes.

### 2.4. Data Preprocessing

During image processing, the first 10 time volumes were removed. Then, the slicing timing was aligned and instances of head motion with a maximum displacement greater than 2 mm in any direction or an angular rotation greater than 2° were excluded from further analyses. The images were then spatially normalized to the Montreal Neurological Institute (MNI) template (resampling voxel size =3 × 3 × 3 mm voxels) and spatially smoothed with a Gaussian kernel of 6 × 6 × 6 mm as the full width at half maximum.

### 2.5. Independent Component Analysis

ICA was performed with GIFT software (Group ICA fMRI toolbox, NITRC, Neuroimaging Tools and Resource Collaboratory, San Diego, CA, USA, https://www.nitrc.org/projects/gift, accessed on 1 June 2020, version 2.0a). The procedure included three steps: (1) principal component analysis to reduce the data at the individual level; (2) application of the InfoMax algorithm to decompose the data into group-independent components; (3) regular back-reconstruction for the time series and spatial maps of the independent components. There is no consensus on the superior algorithm for selecting the optimal number of components. In this study, 30 components were separated from the data for all participants and normalized into Z values. This enabled us to obtain independent spatial activation distribution maps and the corresponding time series, which reflected the measures of intra-FC. Via visual inspection, we identified 16 meaningful resting-state networks (RSNs) as anatomically and functionally classical ICs [34,35,36] (Figure 1): two left frontoparietal networks (LFPN), two right frontoparietal networks (RFPN), three dorsal attention networks (DAN), an executive control network (ECN), two visual networks (VN), three sensorimotor networks (SMN), a default mode network (DMN), and a ventral attention network (VAN).

### 2.6. Statistical Analysis

We used SPM12 software for the data analysis (Statitical Parametric Mapping, University College of London, London, UK, https://www.fil.ion.ucl.ac.uk/spm/, accessed on 30 May 2020, version SPM12). We conducted a single sample *t*-test of the ICs to obtain the masks of the networks in the PA group and HC group, respectively (*p* < 0.001, voxel size > 10, none). Differential brain region connectivity between the groups was identified using an independent two-sample *t*-test. Specifically, we compared the FC values between the PA and HC groups in terms of co-masks (voxel *p* < 0.001, cluster *p* < 0.05, two tailed, Gaussian random field theory correction (GRF)).

The brain areas with significantly different FCs between the two groups were selected as ROIs (regions of interest), and we assessed the relationships between the mean Z-values of these ROIs and clinical characteristics including MoCA score, BDI scores, S-AD scores, T-AI scores, THI score, duration, and pure tone average via Spearman correlation analyses with a threshold of *p* < 0.05. 

## 3. Results

### 3.1. Aberrant Intrinsic Connectivity

Compared with those in the HC group, the VAN, DAN, VN, LFPN, RFPN, SMN, and ECN in the PA group exhibited significant changes in FC (two-sample *t*-test, voxel *p* < 0.001, cluster *p* < 0.05, GRF) (Figure 2). Compared with the HC group, the PA group showed decreased connectivity in the right superior temporal gyrus (STG) within the VAN, right superior parietal lobule (SPG), left calcarine (CAL) and median cingulate and paracingulate gyri (DCG) within the DAN, left cerebellum Crus1 (Crus1) within the VN, left middle occipital gyrus (MOG) within the LFPN, right angular gyrus (ANG) and right supramarginal gyrus (SMG) within the RFPN, left paracentral lobule (PCL) and right superior frontal gyrus (SFG) within the SMN, and increased connectivity in the left cerebellar lobe 6 (lobule 6) within the ECN. Table 2 provides details regarding the brain regions with significant differences in intrinsic FC between the PA and HC groups.

### 3.2. Correlation between FC of RSNs and Clinical Characteristics

The Spearman correlation analysis indicated that the MoCA scores were negatively correlated with the FC values of the left CAL within the DAN (*p* < 0.05, Table 3, Figure 3). The degree of hearing loss was negatively correlated with the FC values of the left PCL within the SMN (*p* < 0.05, Table 4, Figure 3). BDI scores were negatively associated with the FC value of the left Crus1 within the VN (*p* < 0.05, Table 3, Figure 3).

## 4. Discussion

Our data indicate that individuals with bilateral idiopathic tinnitus, hearing loss, and cognitive impairment experienced distinct brain functional reorganization. We found significantly decreased intra-network FC within the VAN, DAN, LFPN, RFPN, VN, and SMN, and increased intra-network FC in the ECN in the PA group. These network alterations provide clues regarding the mechanisms by which tinnitus leads to impaired cognition. Additionally, compared with those in the HC group, in the PA group, FC in the left PCL within the SMN was found to be negatively correlated with the pure tone average, the FC of the left CAL within the DAN was found to be positively correlated with the severity of cognitive disorder, and the FC of Crus 1 within the VN was found to be negatively correlated with BDI score. These indicators could be used as biomarkers to evaluate the severity of cognitive disorders, hearing loss, and depression in patients with tinnitus, hearing loss, and cognitive impairment.

We found no significant differences in the AN between the PA and HC groups, which suggests that AN intra-connectivity was not modified by the experience of tinnitus. Similar results were obtained by fMRI functional studies of chronic tinnitus with hearing loss [13]. A predictive coding model may be used to explain this finding. The model assumes that the sensory system is organized hierarchically, with lower-level sensory information sent to higher-level structures, and higher-level structures sending predictions to the nucleus below, resulting in auditory attention when ascending information is inconsistent with predictions from higher-level structures. The authors proposed that transient lower-level changes are responsible for tinnitus episodes. Specifically, while the experience of auditory stimuli can return to approximately pre-tinnitus levels with habituation, changes in the attentional network maintain the perception of tinnitus. Thus, while we found no abnormal alterations within the AN network, we found such changes in the attention-related networks (VAN, DAN, LFPN, and RFPN) in patients with tinnitus, hearing loss, and cognitive impairment.

We identified decreased functional connectivity of the right STG within the VAN in our patient group. The STG contains the center of the primary auditory cortex and is related to auditory perception [37]. The VAN is reported to be involved in involuntary reorienting following the detection of salient stimuli [24]. The decreased efficiency of the connection between the STG within the VAN in tinnitus patients may make it harder for them to shift their attention to normal acoustic stimuli, which could maintain the perception of tinnitus [38]. An event-related potentials study explored attentional functioning and also found decreased VAN connectivity in tinnitus patients [39]. Therefore, decreased functional connectivity of the right STG within the VAN may be the cause of tinnitus perception.

Compared with the HC group, the PA group had significantly decreased intra-network FC in the DAN, RFPN, and LFPN. These RSNs are considered to form the cognitive control network (CCN), which is involved in a number of functions including top-down modulation, selective spatial attention, working memory, and distracter suppression [40].

The DAN is mainly responsible for top-down attention orientation, and especially exogenous attention orientation [41]. Such attentional processes are primarily initiated by cues about content, the timing of stimuli, and spatial location, and have been particularly well-documented in the visual and spatial domains [42]. The CAL is located in the primary visual cortex and plays a role in the perception and processing of visual stimulation. The SPG is also involved in the visual network and participates in the spontaneous regulation of attention and other functions [43]. In our study, the decreased intranet FC in the right SPG and the left CAL of the DAN in tinnitus patients with hearing loss and cognitive impairment might have reflected disrupted DAN activity, leading to impaired vision-related goal-directed cognition. Schmidt et al. also found that impaired visual information processing in tinnitus patients was associated with abnormal connectivity of the DAN [14]. Additionally, we found decreased intranet FC of the left Crus 1 in the VN. The left Crus 1 is mainly involved in working memory during cognitive processes [44]. The reduced intranet connection within the VN might be related to poor working memory. Furthermore, the DCG is part of the limbic system, which is involved in many functions including executive control, decision making, learning, and emotion [45]. The reduced FC of the right DCG within the DAN may result in poor cognitive performance in tinnitus patients, which is consistent with several structural and functional studies [46,47,48]. Weakened connections between the left CAL, the right SPG, and the right DCG within the DAN and the left Crus 1 within the VN may represent a decline in cognitive abilities, especially visual processing of selective spatial attention in tinnitus patients with hearing loss and cognitive impairment.

We found reduced functional connectivity of the left MOG within the LFPN, the right ANG, the right SFG, and the right SMG within the RFPN in tinnitus patients with hearing loss and cognitive impairment. The PFN plays an important role in higher cognitive control, such as exogenously triggered initiation of control, adjusting control after errors, and moment-to-moment executive control as in repeated rapid task-switching [49,50]. As a lateralized, independent network, the LFPN is primarily responsible for language and the RFPN is in charge of decision-making and cognitive control [51]. The MOG belongs to the visual center of the brain and is also involved in the processing of auditory spatial information [52]. That intranet FC was decreased in the left MOG of the LFPN implies that the tinnitus patients had reduced working memory function. The SFG plays an important role in executive function, emotion processing, and attention [53]. Hyperactivation in the right supramarginal and angular gyrus has been associated with tinnitus [54]. While the FC of these regions was reduced within the RFPN, tinnitus occupies attentional reserves, leaving fewer resources available for cognitive control. This could decrease the intrinsic network activity of the RFPN, and ultimately, diminish top-down cognitive control. Indeed, a previous study reported altered RFPN in tinnitus patients [55].

We found reduced functional connectivity of the left PCL within the SMN in the PA group. The SMN plays a central role in the detection and processing of sensory inputs and the preparation and execution of motor functions. The PCL is the main part of the primary motor cortex [56]. Panouillères proposed the auditory-motor decline hypothesis, which states that defects in the auditory system lead to reduced recruitment of the motor system during speech processing in hearing loss patients [57]. In our study, we included the MoCA as a measure of motor function, as poorer performance of tinnitus patients with hearing loss and cognitive impairment in this task may be associated with a disruption in the SMN. Tinnitus has been speculated to be a proprioceptive illusion associated with widespread emotional and somatosensory dysfunction, indicating that the observed changes in sensorimotor activity might be primarily associated with hearing loss [58]. A fMRI study also demonstrated disrupted connectivity involving the SMN in tinnitus patients with hearing loss [59]. Similarly, we found that the FC values of the PCL within the SMN were correlated with the hearing loss threshold.

We found enhanced FC of lobule 6 within the ECN in the PA group. This increased cerebellar-brain FC might be related to the perception of tinnitus and tinnitus-related distress [60]. The ECN is involved in several higher-level cognitive tasks, mostly related to activity inhibition and emotion, and it is also involved in sustained attention and working memory [50]. The left lobule VI is considered part of the neural circuit associated with anxiety and fear [60,61]. Therefore, abnormal ECN connections to the cerebellum may be related to tinnitus-related distress. Additionally, the FC value of the left Crus 1 with the VN was negatively correlated with BDI scores. As mentioned, the observed changes in the VN and left Crus I were mainly related to working memory in tinnitus patients. Depression is thought to have a negative impact on information processing [18]. Thus, low mood might play an important role in the development of cognitive impairment in tinnitus patients with hearing loss and cognitive impairment. Our data further confirm the importance of the cerebellum in tinnitus cognition and mood.

In summary, bilateral tinnitus patients with hearing loss and cognitive impairment showed reduced connectivity within networks, including the top-down control networks (DAN, RFPN, LFPN), VN, and SMN. This suggests that cognitive impairment in these patients is mainly related to cognitive control-related tasks, sensory abilities, and motor abilities. The observed alterations in the cerebellar-brain connection also suggest that emotion plays an indispensable role in the development of cognitive impairment in this patient group. Additionally, we found that FC values of the left CAL within the DAN were negatively associated with MoCA scores, indicating that this could be a biomarker for evaluating the severity of cognitive impairment in tinnitus patients with hearing loss and cognitive impairment.

## 5. Limitations

This study has several shortcomings. First, the sample size of our study was relatively small. Second, we did not consider the severity of cognitive impairment in terms of brain changes. Third, we only enrolled patients with bilateral tinnitus and hearing loss and did not examine brain changes in unilateral tinnitus patients and tinnitus patients with normal hearing. Further studies are needed to explore the mechanisms of tinnitus with different clinical characteristics and larger sample populations.

## 6. Conclusions

Resting-state functional MRI ICA revealed no significant changes in the AN in tinnitus patients with cognitive impairment. We observed disruptions in the VAN, which may be involved in the generation and maintenance of tinnitus. Alterations in several networks related to cognitive control (DAN, LFPN, and RFPN), visual processing (VN), and sensor-motor function (SMN) were correlated with poor cognitive processing. Enhanced cerebellar-brain connectivity indicates that emotion plays a major role in the development of cognitive impairment in tinnitus patients. The FC strength of the left CAL within the DAN may serve as an imaging biomarker of cognitive impairment in tinnitus patients.

## Figures and Tables

**Figure 1 brainsci-12-01049-f001:**
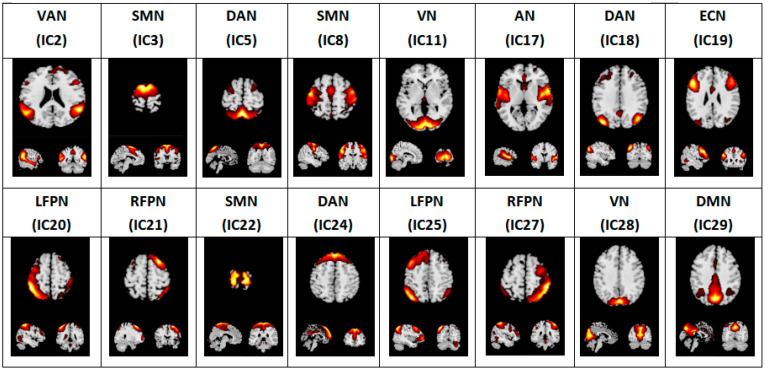
Sixteen meaningful resting-state networks identified by ICA. IC: independent component; LFPN: left frontoparietal network; RFPN: right frontoparietal network; DAN: dorsal attention network; ECN: executive control network; VN: visual network; SMN: sensorimotor network; AN: auditory network; DMN: default mode network; VAN: ventral attention network.

**Figure 2 brainsci-12-01049-f002:**
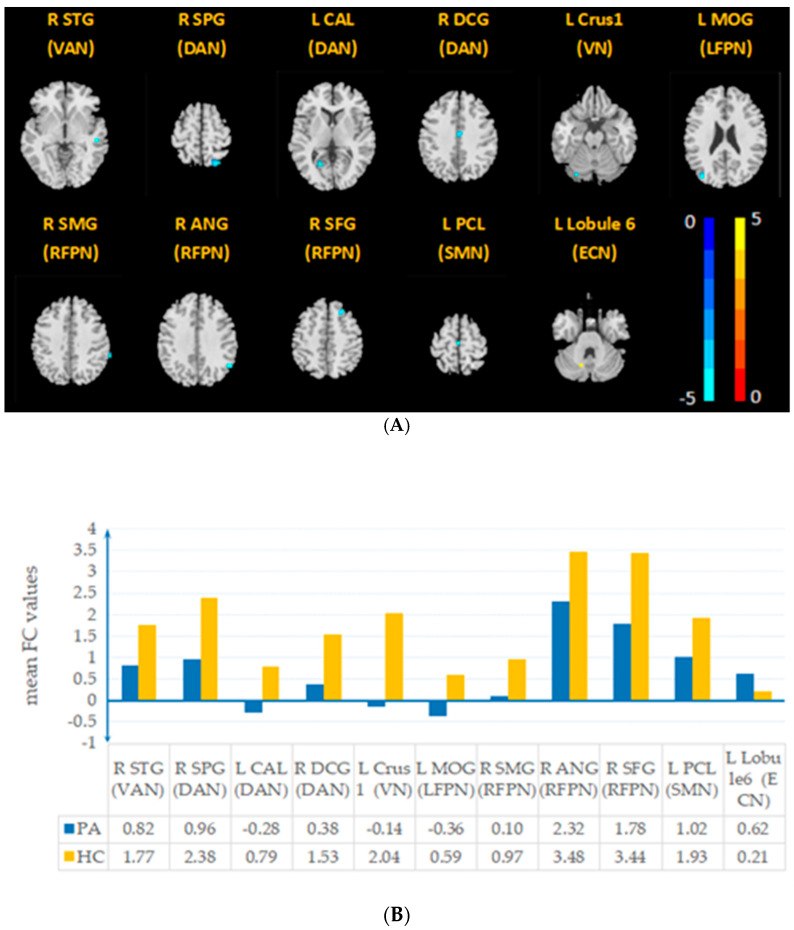
Brain regions within RSNs were significantly different between the PA and HC groups (**A**). The X-axis represents the brain regions contained in the RSNs, and the Y-axis represents the mean value of the intrinsic FC signal (**B**). The detailed regions can be found in Table 2.

**Figure 3 brainsci-12-01049-f003:**
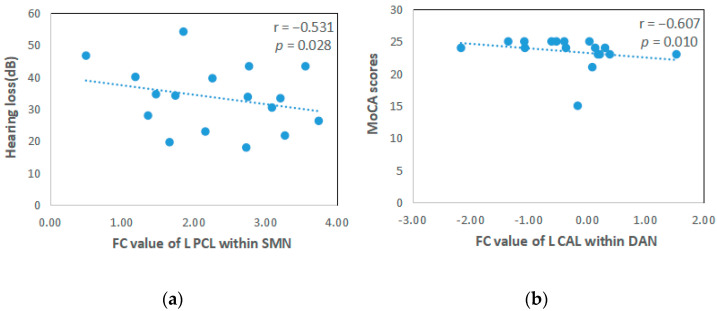
Correlations between clinical characteristics and FC values of abnormal brain regions in patients with bilateral idiopathic tinnitus and hearing loss with cognitive impairment. Hearing thresholds were negatively correlated with FC values of the left paracentral lobule within the paracentral lobule (r = −0.531, *p* = 0.028). MoCA scores were negatively correlated with FC values of the left calcarine within the dorsal attention network (r = −0.607, *p* = 0.010). BDI scores were negatively correlated with FC values of the left Crus 1 within the visual network (r = −0.549, *p* = 0.022).

**Table 1 brainsci-12-01049-t001:** Demographics and clinical characteristics of the PA and HC groups.

Demographic (Mean ± SD)	PA	HC	T-Value	*p*-Value
Gender (M/F)	10/7	7/14	2.404	0.121 ^a^
Age (years)	54.76 ± 8.92	51.24 ± 6.33	1.424	0.163 ^b^
Education (years)	11.82 ± 2.19	12.90 ± 2.81	−1.299	0.202 ^b^
Handedness(right/left)	17/0	21/0	N/A	1.000 ^a^
Duration (years)	2.61 ± 2.50			
THI	48.12 ± 30.50			
Pure Tone Average (dB)	33.50 ± 10.18			
MoCA	23.41 ± 2.43			
BDI	11.53 ± 11.41			
S-AI	38.12 ± 11.68			
T-AI	37.47 ± 9.29			

Data are presented as mean ± SD; PA: patient group; HC: healthy controls; THI: Tinnitus Handicap Inventory; MoCA: Montreal Cognitive Assessment; BDI: Beck Depression Inventory; S-AI: State Anxiety Inventory; T-AI: Trait Anxiety Inventory; N/A: not applicable. ^a^ chi-square test; ^b^ two-sample *t*-test.

**Table 2 brainsci-12-01049-t002:** Brain regions with significant differences in intrinsic functional connectivity between the PA and HC groups.

	RSN	Brain Region	Cluster Size, mm³	Peak MNI, mm	Peak *t* Value
				x y z	
PA < NC	VAN (IC2)	R STG	16	51 −27 −3	−4.4865
	DAN (IC5)	R SPG	69	24 −69 63	−6.9585
	DAN (IC18)	L CAL	14	−18 −63 6	−4.4156
	DAN (IC24)	R DCG	16	3 −12 39	−4.9841
	VN (IC11)	L Crus1	10	−27 −81 −21	−5.2585
	LFPN (IC20)	L MOG	11	−39 −81 24	−5.7306
	RFPN (IC22)	R ANG	12	54 −57 36	−4.8083
	RFPN(IC27)	R SMG	11	66 −39 39	−4.7568
	SMN (IC21)	L PCL	12	−3 −18 66	−4.2251
		R SFG	20	24 27 51	−5.0179
PA > NC	ECN (IC19)	L Lobule 6	10	−15 −63 −30	4.7252

PA: patient group; HC: healthy controls; MNI: Montreal Neurologic Institute; R: right; L: left; VAN: ventral attention network; DAN: dorsal attention network; VN: visual network; LFPN: left fronto-parietal network; RFPN: right fronto-parietal network; SMN: sensorimotor network; ECN: executive control network; STG: superior temporal gyrus; SPG: superior parietal lobule; CAL: calcarine; DCG: median cingulate and paracingulate gyri; Crus 1: cerebellum Crus 1; MOG: middle occipital gyrus; SMG: supramarginal gyrus; ANG: angular gyrus; SFG: superior frontal gyrus; PCL: paracentral lobule; lobule 6, cerebellar lobe 6.

**Table 3 brainsci-12-01049-t003:** Correlations between clinical characteristics and FC values of abnormal brain regions.

Brian Regions		MoCA Scores		BDI Scores		S-AI Scores		T-AI Scores	
		r	*p*	r	*p*	r	*p*	r	*p*
VAN	R STG	0.093	0.722	0.161	0.537	0.200	0.441	0.198	0.446
	R SPG	0.474	0.055	−0.137	0.601	0.357	0.159	0.305	0.234
DAN	L CAL	−0.607 *	0.010	0.000	1.000	0.301	0.241	0.152	0.559
	R DCG	0.252	0.330	−0.241	0.351	−0.430	0.085	−0.036	0.892
VN	L Crus1	0.071	0.787	−0.549 *	0.022	−0.151	0.563	−0.127	0.626
LFPN	L MOG	0.406	0.106	−0.233	0.369	−0.247	0.339	−0.329	0.197
	R SMG	−0.405	0.107	0.134	0.608	−0.108	0.680	0.181	0.488
RFPN	R ANG	0.324	0.204	−0.165	0.527	−0.076	0.772	0.042	0.873
	R SFG	0.080	0.759	−0.235	0.364	−0.377	0.136	−0.479	0.052
SMN	L PCL	0.012	0.965	−0.120	0.646	−0.198	0.445	−0.271	0.293
ECN	L Lobule_6	−0.014	0.957	0.085	0.746	0.080	0.761	−0.004	0.989

MoCA: Montreal Cognitive Assessment; BDI: Beck Depression Inventory; S-AI: State Anxiety Inventory; T-AI: Trait Anxiety Inventory; R: right; L: left; VAN: ventral attention network; DAN: dorsal attention network; VN: visual network; LFPN: left fronto-parietal network; RFPN: right fronto-parietal network; SMN: sensorimotor network; ECN: executive control network; STG: superior temporal gyrus; SPG: superior parietal lobule; CAL: calcarine; DCG: median cingulate and paracingulate gyri; Crus 1: cerebellum Crus1; MOG: middle occipital gyrus; SMG: supramarginal gyrus; ANG: angular gyrus; SFG: superior frontal gyrus; PCL: paracentral lobule; lobule 6, cerebellar lobe 6. * *p* < 0.05.

**Table 4 brainsci-12-01049-t004:** Correlations between clinical characteristics and FC values of abnormal brain regions.

		Hearing		THI		Duration		Tinnitus-Hz		Tinnitus-dB	
		r	*p*	r	*p*	r	*p*	r	*p*	r	*p*
VAN	R STG	0.012	0.963	0.120	0.646	0.402	0.123	0.189	0.484	0.041	0.879
	R SPG	0.166	0.525	0.100	0.701	0.420	0.106	−0.048	0.859	0.482	0.059
DAN	L CAL	−0.426	0.089	−0.125	0.633	−0.394	0.131	−0.045	0.868	−0.279	0.295
	R DCG	−0.013	0.959	−0.184	0.480	0.321	0.225	−0.069	0.798	−0.155	0.566
VN	L Crus 1	−0.015	0.955	−0.416	0.097	−0.039	0.887	−0.041	0.881	0.152	0.573
LFPN	L MOG	0.309	0.227	−0.219	0.397	0.450	0.080	0.289	0.277	0.371	0.157
	R SMG	0.262	0.309	0.020	0.940	−0.146	0.590	−0.486	0.056	−0.167	0.537
RFPN	R ANG	0.152	0.560	−0.140	0.593	0.202	0.452	−0.470	0.066	0.021	0.939
	R SFG	−0.235	0.363	−0.279	0.277	−0.283	0.289	−0.056	0.837	−0.131	0.627
SMN	L PCL	−0.531	0.028 *	−0.055	0.833	−0.107	0.693	−0.101	0.709	−0.378	0.149
ECN	L Lobule_6	−0.042	0.874	0.456	0.066	−0.028	0.917	−0.133	0.624	−0.186	0.490

THI: Tinnitus Handicap Inventory; R: right; L: left; VAN: ventral attention network; DAN: dorsal attention network; VN: visual network; LFPN: left fronto-parietal network; RFPN: right fronto-parietal network; SMN: sensorimotor network; ECN: executive control network; STG: superior temporal gyrus; SPG: superior parietal lobule; CAL: calcarine; DCG: median cingulate and paracingulate gyri; Crus 1: cerebellum Crus1; MOG: middle occipital gyrus; SMG: supramarginal gyrus; ANG: angular gyrus; SFG: superior frontal gyrus; PCL: paracentral lobule; lobule 6, cerebellar lobe 6. * *p* < 0.05.

## Data Availability

The data presented in this study are available on request from the corresponding author. The data are not publicly available due to data privacy.

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
