# Peer review of "Intrinsic Network Changes in Bilateral Tinnitus Patients with Cognitive Impairment: A Resting-State Functional MRI Study"

_brainsci, 2022, doi:10.3390/brainsci12081049_

Round 1

Reviewer 1 Report

Dear Ladies and Gentlemen, Dear Journal-Team,

the manuscript 'Intrinsic network changes in bilateral tinnitus patients with cognitive impairment: A resting-state functional MRI study' investigates the functional connectivity of the brain networks. It is well written. Tables and Figures are sufficient.

a) Was a correlation between the tinnitus, the hearing loss side, and the brain side possible?

b) Please define the auditory network (AN), for example in the introduction. As changes have been found within the angular and supramarginal gyri in the study and literature.

c) Please clarify the statement in the Limitations section, that the severity of cognitive impairment was not considered in terms of brain changes. As the Montreal Cognitive Assessment was correlated with the functional connectivity values.

d) Please explain every abbreviation, when first mentioned: MoCA scores (Montreal Cognitive Assessment) in the abstract, ROI (region of interest) in statistical analysis, explain MNI (Montreal Neurologic Institute) in Table 2, explain NC (normal connectivity) in Table 2, explain AN (auditory network) not only in Figure 1. Check Table 1 for the abbreviation S-AD in the first column. Why was the abbreviation DCG used for the median cingulate gyrus? 

e) Extend the information of the software used with the explanation of the abbreviation and the main residenece of the manufacturer or inventor: GIFT (Group ICA fMRI toolbox, NITRC, Neuroimaging Tools & Resource Collaboratory, US), SPM (Statitical Parametric Mapping, University College of London).

f) Please check the References for accuracy according to the Journal Style Guidelines. Mention at the end of the reference the language, if the article is not in English. Check Reference 31 for punctuation and parentheses. Check Reference 50 for spacing within 'Brain's'.

g) Conclusion, line 1, please change to: 'in the AN'.

Sincerely,

Reviewer 2 Report

The main finding of the study is the investigation of intrinsic connectivity in several brain networks in tinnitus patients with hearing loss and cognitive impariment.

I suggest only minor corrections.

Title: Intrinsic Network Changes in Bilateral tinnitus patients with hearing loss and cognitive impairment: a resting-state functional MRI study

Did you exclude patients with acoustic trauma and vestibular schwannoma?

line 57: De Ridder

line 97: bilateral sensorineural hearing loss above 25 dB HL: on high frequencies? on 2 or more frequencies?

Table 1: the term THI in the legend is for Tinnitus Handicap Inventory; S-AD is not correct (S-AI)

Table 4: the term THI in the legend is for Tinnitus Handicap Inventory

How many patients were with mild/moderate/severe hearing loss based on the audiological evaluation?

In the limitations of the study you can add that you did not examine tinnitus patients with normal hearing

Author Response

Dear Reviewer,

Thank you very much for your time involved in reviewing the manuscript. We appreciate your clear and detailed feedback and hope that the explanation has fully addressed all of your concerns. In the remainder of this letter, we discuss each of your comments individually along with our corresponding responses.

Point 1: Did you exclude patients with acoustic trauma and vestibular schwannoma?

Response 1: Yes, we excluded patients with a history of ear trauma and a history of medical conditions other than tinnitus and hearing loss.

Point 2: line 57: De Ridder

Response 2: Thanks to the reviewer for your reminder, it has been changed.

Point 3: line 97: bilateral sensorineural hearing loss above 25 dB HL: on high frequencies? on 2 or more frequencies?

Response 3: In this experiment, hearing loss was defined as a hearing threshold greater than 25 dB per side in any of six frequencies (0.25, 0.5, 1, 2, 4, and 8 kHz). We explained further in the section on audiological examination.

Point 4: Table 1: the term THI in the legend is for Tinnitus Handicap Inventory; S-AD is not correct (S-AI)

Response 4: Thanks for reviewer’s reminder, we have corrected it.

Point 5: Table 4: the term THI in the legend is for Tinnitus Handicap Inventory

Response 5: Thanks for reviewer’s reminder, we have corrected it.

Point 6: How many patients were with mild/moderate/severe hearing loss based on the audiological evaluation?

Response 6: According to the WHO 2021 version of the classification criteria for hearing loss, on the better side the mean hearing threshold at 500, 1k, 2k and 4k rating rates, <20 is normal, 20-35 dB is mild hearing loss, and 35-50 is moderate. Of the 17 patients, 5 had normal hearing in the better ear, 5 had mild hearing loss, and 7 had moderate loss.

Point 7: In the limitations of the study you can add that you did not examine tinnitus patients with normal hearing.

Response 7: Thanks to the reviewer's suggestion, we have added the point at limitation.

Finally, thank you very much for your suggestions, we have invited native English speakers ( Sydney Koke, MFA, from Liwen Bianji (Edanz) (www.liwenbianji.cn) ) to revise the language grammar, and we will continue to study English writing to improve our writing.
